# E2F-dependent transcription determines replication capacity and S phase length

Betheney R. Pennycook [1,2], Eva Vesela[1], Silvia Peripolli[1], Tanya Singh[1], Alexis R. Barr [2,3], Cosetta Bertoli [1✉] & Robertus A. M. de Bruin [1,4✉]

DNA replication timing is tightly regulated during S-phase. S-phase length is determined by DNA synthesis rate, which depends on the number of active replication forks and their velocity. Here, we show that E2F-dependent transcription, through E2F6, determines the replication capacity of a cell, defined as the maximal amount of DNA a cell can synthesise per unit time during S-phase. Increasing or decreasing E2F-dependent transcription during S-phase increases or decreases replication capacity, and thereby replication rates, thus shortening or lengthening S-phase, respectively. The changes in replication rate occur mainly through changes in fork speed without affecting the number of active forks. An increase in fork speed does not induce replication stress directly, but increases DNA damage over time causing cell cycle arrest. Thus, E2F-dependent transcription determines the DNA replication capacity of a cell, which affects the replication rate, controlling the time it takes to duplicate the genome and complete S-phase.

[1] MRC Laboratory for Molecular Cell Biology, University College London, Gower street, London WC1E 6BT, UK. [2] MRC London Institute of Medical Science Hammersmith Hospital Campus, Du Cane Road, London W12 0NN, UK. [3] Institute of Clinical Sciences, Imperial College London, London W12 0NN, UK. [4] UCL Cancer Institute, University College London, London WC1E 6BT, UK. ✉email: c.bertoli@ucl.ac.uk; r.debruin@ucl.ac.uk

One of the key events for the regulation of DNA replication is the temporal separation of replication initiation into licensing, that is helicase loading onto DNA, and firing, i.e. conversion to two active replication forks[1–5]. While all origins are licensed only a limited number of these fire during S phase, with the majority of licensed origins remaining dormant[6]. Origin firing is highly regulated through a temporal programme of replication initiation during S phase which is tightly related to chromatin structure, nuclear architecture and transcriptional regulation[7]. The temporal programme of origin firing ensures that only a limited number of replication forks are active at any one time during S phase. In addition, the presence of dormant origins allows origin usage to change according to cellular context, providing plasticity to the genome duplication process.

The time it takes to complete genome duplication, and therefore S-phase length, depends on the DNA synthesis rate. The DNA synthesis rate is determined by the number of active replication forks and replication fork speed. However, there is a clear negative correlation between replication fork speed and the rate of replication initiation[8,9] (Fig. 1a). When initiation occurs less frequently, fork speed is faster, whereas a slower fork rate correlates with more frequent initiation[5,9,10]. The inverse correlation between replication initiation and replication fork speed suggests that the amount of DNA a cell can synthesise per unit time, the replication capacity, is limited. We speculate that limiting the replication capacity of a cell would provide an elegant mechanism to regulate the global rate of replication during S phase, largely independent of the number of active replication forks. It would ensure timely completion of genome duplication and prevent potentially harmful alterations in fork speed[11]. However, whether a cell's replication capacity is controlled during S phase and if so what the underlying mechanism could be is currently unknown.

G1/S transcription plays a key role in S-phase entry, coordinating replication with cell-cycle progression (Fig. 1b). G1/S transcription in mammalian cells depends on the E2F family of transcription factors (E2F1-E2F8) and their co-regulators the pocket proteins (pRb, p107 and p130)[12]. The central role of E2F-dependent transcription in driving replication initiation through the expression of proteins required for licensing and firing of origins is well-established[13–15]. In addition, E2F-dependent transcription is required for the expression of many proteins involved in DNA synthesis, DNA repair and cell-cycle progression during the S phase of the cell cycle. Peak transcription of individual G1/S target genes is tightly linked to their function during the G1 to S transition. Cyclin E, which is required early during the transition to increase CDK activity, shows an early expression profile in late G1 phase, while RRM2, needed during DNA replication, is expressed throughout S phase[12,16,17]. Among the late expressed genes of the G1–S regulon is the E2F family repressor E2F6[18,19]. During S phase, E2F-dependent transcription is inactivated through negative feedback loops in which E2F6 plays a key role[19,20]. It is probable that the late expression of E2F6 allows for the expression of genes needed for replication before E2F6-dependent repression of G1/S transcription is initiated[12,18,19,21,22]. This mechanism allows genes needed for replication to be sufficiently expressed during S phase.

## Results

### E2F6 knockdown significantly speeds up S-phase progression.
Overexpression or depletion of individual E2F targets that encode for replication factors affect replication dynamics and cause genomic instability[23–25]. However, it is not known how the deregulation of the entire E2F regulon during S phase affects replication dynamics, cell-cycle progression and genome stability.

To test this, we first investigated if an overall increase in E2F-dependent transcription during S phase affects the timely duplication of the genome. We have previously established that E2F6 knockdown maintains E2F-dependent transcription at a high level during S phase in T98G cells[20]. We first asked if sustained E2F-dependent transcription during S phase has consequences for S-phase progression. T98G cells were transfected with siRNA targeting E2F6 before synchronisation by serum starvation. Cells were released from starvation into a mitotic arrest via Nocodazole block, and cell-cycle progression was followed through analysis of DNA content by flow cytometry (Fig. 1c, Supplementary Fig. 1a, b). As expected, the timing of S-phase entry is largely unaffected in the siE2F6 cells. However, E2F6 knockdown significantly speeds up S-phase progression with a higher percentage of E2F6-depleted cells completing S phase at 26 and 28 h, 48.7% and 63.5%, respectively, compared with 14.1% and 27.5% in control cells. This indicates a decrease in S-phase length in siE2F6-treated cells. To further confirm this observation, we took a single-cell live imaging approach. Human RPE1 hTERT cells expressing endogenously tagged mRuby-PCNA and p21-GFP were imaged and the intensity and spatial distribution of mRuby-PCNA was used to determine cell-cycle timing in single cells following E2F6 knockdown (Fig. 1d)[26–28]. This allowed us to monitor cell-cycle progression in single cells in an asynchronous unperturbed cell population. Live-cell tracking and quantification of PCNA pattern and intensity confirms that E2F6 knockdown decreases the length of S phase (Fig. 1e, f, Supplementary Fig. 1c, d). siE2F6-treated cells display a significantly and consistently shorter S-phase length from the first cell cycle following knockdown across multiple cell generations (Fig. 1g, Supplementary Fig. 1d). This is not coupled with a significant change in total cell-cycle length (Supplementary Fig. 1e). Together, this data indicate that the primary effect of maintained E2F-dependent transcription during S phase on cell-cycle progression is a decrease in S-phase length.

### Sustained E2F-dependent transcription in S phase increases replication fork speed.
We next sought to determine how DNA replication dynamics are affected by E2F6 depletion to allow for a faster S phase. To assay global DNA synthesis rate, we pulsed labelled cells for 30 min with EdU and measured its incorporation by quantitative immunofluorescence (Fig. 2a and Supplementary 2a). Cells treated with siE2F6 display a small but significant increase in total EdU incorporation per cell compared with control cells, indicating an increase in DNA synthesis rate. DNA synthesis rate is determined by the number of active replication forks and their speed. Therefore, an increase in the DNA synthesis rate as a result of E2F6 knockdown can derive from either an increase in the number of active forks or an increase in fork speed or both. Flow cytometry and western blot were used to assess levels of chromatin-bound MCMs and PCNA as a measure of origin licensing and firing, respectively[29]. Interestingly E2F6 knockdown does not result in a significant difference in chromatin-bound protein levels of either MCM2 and 7 or PCNA (Fig. 2b and Supplementary Fig. 2b–d), indicating that the number of active replication forks is not significantly affected upon E2F6 knockdown. In agreement with this finding, the amount of origin firing observed in the DNA fibre spreads did not change upon E2F6 depletion (Supplementary Fig. 2e). Conversely, when origin firing was reduced by treatment with CDK inhibitors we detected a decrease in chromatin-bound PCNA and in global DNA synthesis in our system (Supplementary Fig. 2f). This indicates a reduction in the number of active forks upon CDK inhibition, which is in line with previously reported results in other cell systems[30–34]. Next we tested if E2F6 knockdown affects replication fork speed as assayed by DNA fibre analysis.

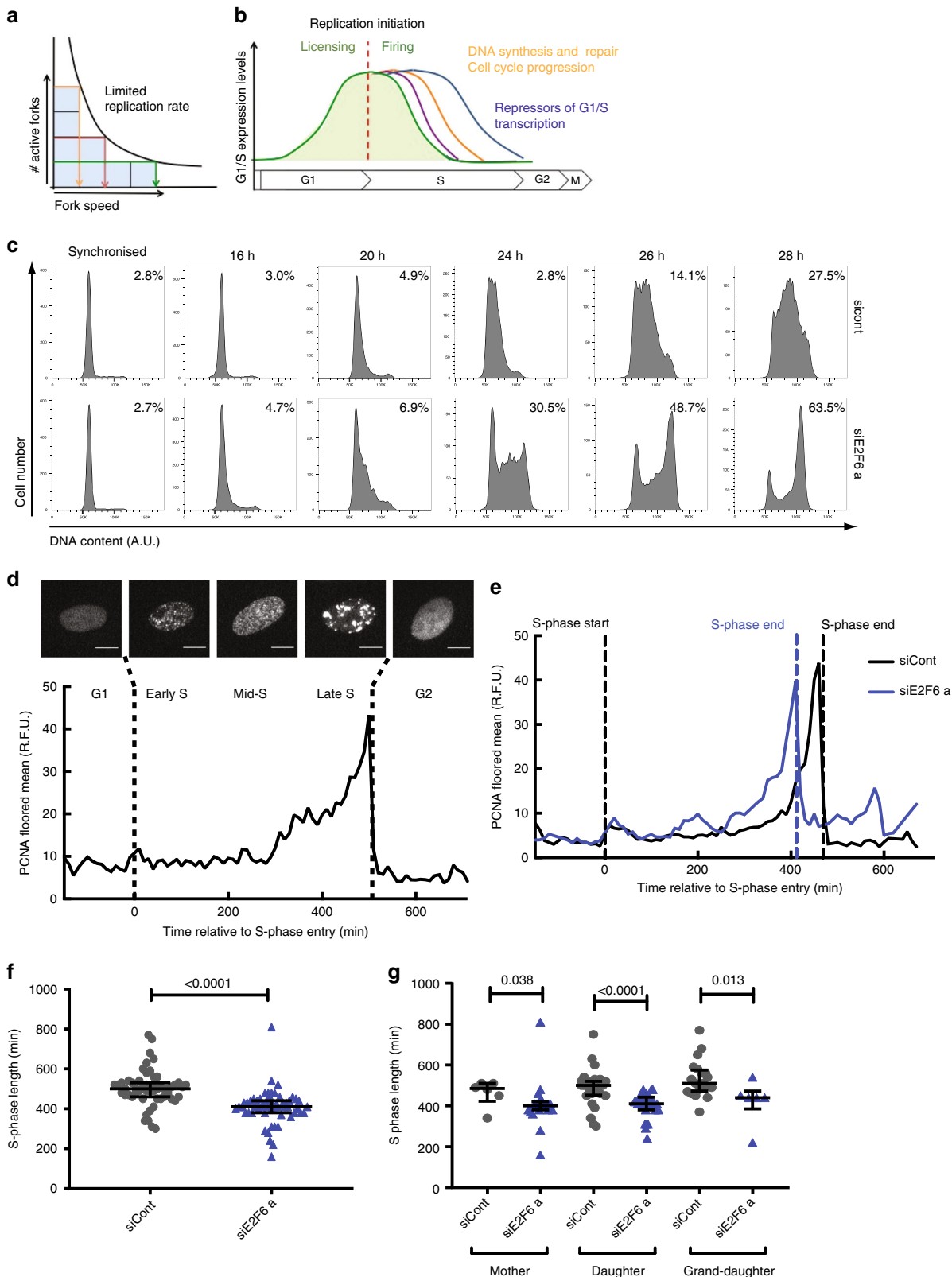

E2F6 knockdown shifts the distribution towards longer fibre track length, indicative of an increase in replication fork speed (Fig. 2c, Supplementary Fig. 2g). An increase in replication fork speed has been reported before, but only as a consequence of a decrease in replication origin activation resulting in under-replication[8,9,34]. Experimentally reducing the number of active forks by CDK inhibition is also coupled to an increase in DNA replication fork speed (Supplementary Fig. 2h). However, depletion of E2F6 is distinctly different in that it appears to increase replication fork speed without reducing origin activity. Based on these results we conclude that sustained E2F-dependent transcription in S phase results in an increase in replication fork speed, thus increasing the overall DNA synthesis rate without any detectable change in origin licensing or firing.

**Fig. 1 Maintaining E2F-dependent transcription in S phase decreases S-phase length. a** Schematic of the negative correlation between the number of active replication forks (# active forks) and the speed at which these travel (fork speed). Shaded area indicates the largely similar global DNA synthesis rates (four squares) for high (yellow), mid (red) and low (green) numbers of active forks corresponding to low, mid and high fork speeds, respectively. **b** Genes regulated as part of the G1/S cell-cycle transcriptional regulon encode for proteins with a central role in the regulation of replication initiation, DNA synthesis and repair, cell-cycle progression and inactivation of G1/S transcription. Individual E2F targets (coloured lines) display slightly different transcriptional profiles, which is thought to be closely linked to their function during the G1 and S phase of the cell cycle. **c** T98G cells were transfected and synchronised in G1 phase by serum starvation, re-transfected and released in serum. Nocodazole was added at 16 h post release at 100 ng/ml. DNA was quantified by flow cytometry at the indicated times following release and transfection with control (LacZ) or E2F6 a siRNA. Inset percentage reflects cells in G2/M phase based on DNA staining in one of three experimental repeats. **d** Images and quantification of mRuby-PCNA levels (floored mean) over time from automated image analysis of a RPE1 hTERT cell expressing endogenously tagged p21-GFP and mRuby-PCNA which was used for cell-cycle stage classification. Scale bar represents 10 μm. **e** mRuby-PCNA levels (floored mean) over time in single representative cells treated with control (black) or E2F6 (blue) siRNAs. **f** Mean S-phase length in single cells plotted as an average across three cell cycles and within each imaged cycle, (**g**). **d**–**g** siCont $n =$ 55 cells ($n = 6, 18, 19$ mother, daughter and grand-daughter cells, respectively), siE2F6a $n = 48$ ($n = 28, 23, 30$). Mean with SD is shown. Significance was determined using a Kruskal–Wallis test (**f**, **g**).

**Sustained E2F-dependent transcription increases replication capacity.** Many genes involved in DNA replication are targets of E2F-dependent G1/S transcription and their expression is limited by E2F6 in late S phase (Supplementary Fig. 3a)[12,19,20]. Our single-cell live imaging data of the well-established E2F target PCNA[35], endogenously tagged with mRuby, confirms published data that E2F6 knockdown mainly causes an increase in the expression of E2F target genes during mid and late S phase[18–20,36] (Fig. 3a). Our data suggests that maintaining the expression of these genes, through E2F6 knockdown, allows for higher fork speeds increasing the overall DNA synthesis rate. This suggests that E2F-dependent transcription controls the amount of DNA a cell can synthesis per unit time in S phase, defined as replication capacity, which we speculate is likely through regulating the expression of limiting factors for DNA replication. So if E2F-dependent transcription limits the replication capacity of a cell, E2F6 knockdown should mainly increase replication capacity in mid and late S phase. Since the increase in replication capacity allows for an increase in fork speed we analysed replication fork speed in synchronized cells during early and middle/late S phase. RPE1 cells were synchronised by contact inhibition to enrich the population of cells for those entering the first S phase following E2F6 knockdown (Supplementary Fig. 3b). While in early S phase the distribution of replication fork track lengths is similar between control and E2F6-depleted samples, in middle/late S phase a significant increase in replication track length can be observed upon E2F6 depletion compared with control (Fig. 3b, Supplementary Fig. 3c), and this is also true for T98G cells (Supplementary Fig. 3d). In addition to this, an increase in the G2/M percentage of cells at 24 h following release in siE2F6-treated cells, supports our hypothesis that there is a decrease in S-phase length (Supplementary Fig. 3e). This is not coupled with a decrease in origin firing (Supplementary Fig. 3f). Together, this indicates that maintained E2F-dependent transcription during mid/late S phase by E2F6 knockdown provides an increase in replication capacity resulting in an increase in replication fork speed.

**A decrease in E2F-dependent transcription in S phase decreases replication capacity.** To further test the hypothesis that the extent of E2F-dependent transcription during S phase determines the replicative capacity of cells, we overexpressed E2F6 during middle/late S phase, thus decreasing E2F transcription, and evaluated replication fork speed and global replication rate as above. In agreement with the above data, overexpression of E2F6 decreases replication fork speed (Fig. 3c), and reduces the overall quantity of DNA synthesis (Fig. 3d). These data show that changes in the levels of E2F-dependent transcription during S phase affect the replication capacity of a cell, which influences the

speed at which replication forks travel without changes to the number of active replication forks.

**Sustained E2F-dependent transcription over time causing cell-cycle arrest.** Recent evidence has suggested that increased replication fork speed, as observed in siE2F6-treated cells, can cause genomic instability[11]. We therefore tested the levels of DNA damage checkpoint activation, via γH2AX intensity on chromatin, upon E2F6 depletion in synchronized cells. Surprisingly, we do not observe an increase in γH2AX intensity at any timepoint during the first S phase following E2F6 knockdown, suggesting that in our system an increased fork speed does not cause DNA damage directly (Fig. 4a and Supplementary Fig. 4a). However, we did observe a small but significant increase in γH2AX chromatin staining, and in the number of cells showing at least one γH2AX focus, in siE2F6-treated cells compared with control cells during the second cell cycle (41 h) (Fig. 4a, Supplementary Fig. 4b). Since low levels of replication stress have been shown to cause a DNA damage response in the next G1 phase[26,37–39], we tested if E2F6-depleted cells are arrested in the next G1. Single-cell time-lapse analysis, using quantification of PCNA pattern and intensity as shown in Fig. 1d, shows a significant increase in the proportion of arrested cells and in G1 length in cells treated with siE2F6 (Fig. 4b, c, Supplementary Fig. 4c, d). In the same setting we analysed DNA damage checkpoint activation by measuring the levels of the CDK inhibitor p21 over time throughout multiple generations of cells. Average levels of p21 per cell appear higher upon E2F6 depletion, confirming checkpoint activation (Fig. 4d, Supplementary Fig. 4e). Importantly, the levels of p21 protein in single cells increase following the first division and remain significantly increased compared with mother cells. This suggests that cells continue to cycle despite high levels of p21 and the presence of DNA damage. Interestingly, we did not observe a decrease in PCNA and MCM7 on chromatin in the second cell cycle (Supplementary Fig. 4f), suggesting there is no reduction in licensing in these cells.

## Discussion

We propose a DNA replication capacity model, whereby controlling the maximal amount of DNA a cell that can synthesise per unit time during the S phase provides a mechanism to determine DNA synthesis rates and S-phase length that is largely independent of replication-initiation events. Our data support a model where E2F transcription—which regulates the expression of replication factors—controls the replication capacity of a cell throughout S phase (Fig. 4e). E2F transcription does not respond to replication-initiation events, but sets the replication rate of a cell independently of the amount of replication-initiation events. E2F-dependent replication capacity control during S phase would provide a mechanism where fluctuations in replication-initiation

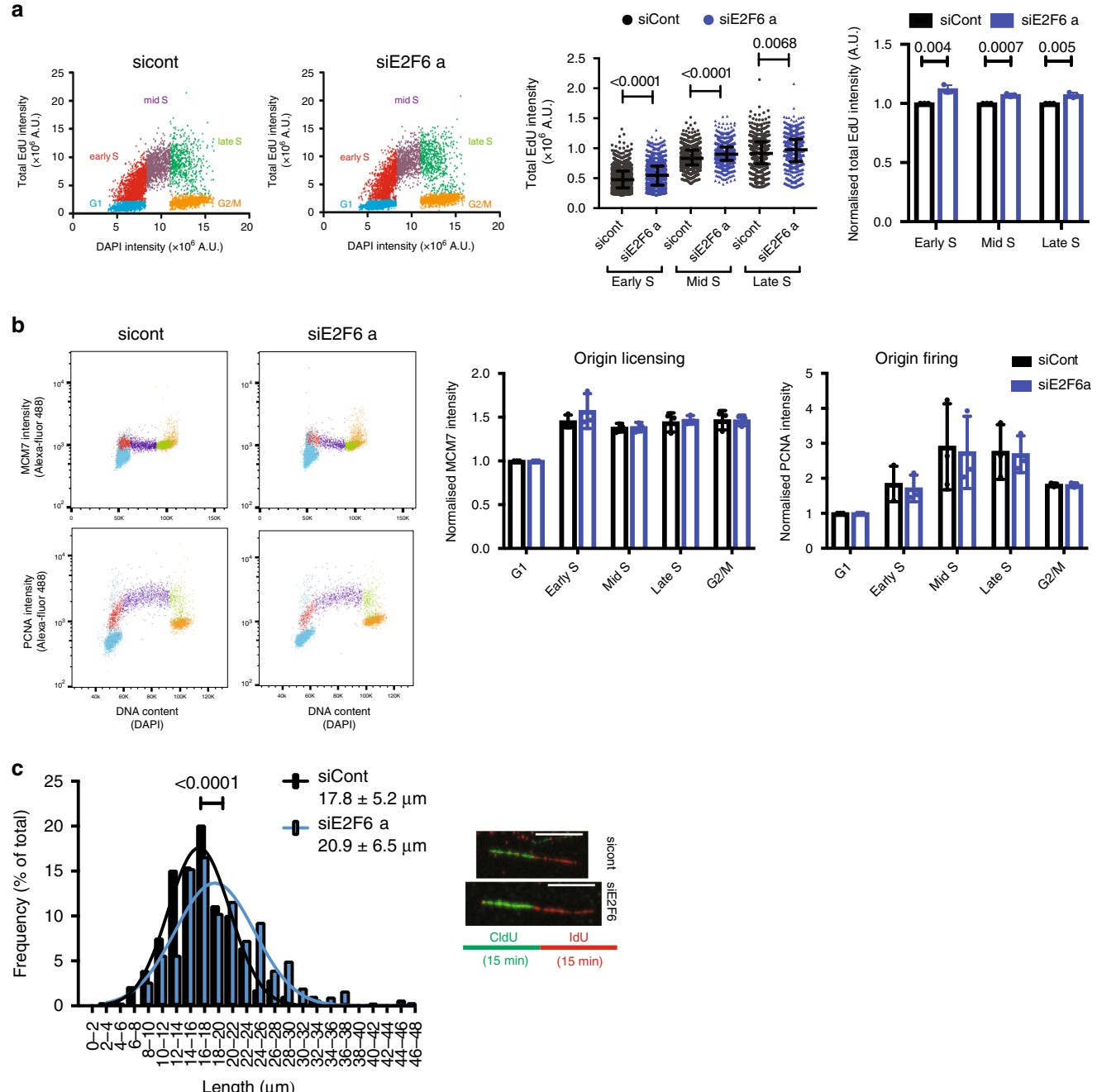

**Fig. 2 Maintained E2F-dependent transcription in S phase increases the replication capacity of cells. a** Asynchronous RPE1 cells were reverse transfected and 24 h later pulsed with EdU for 30 min before fixation and immunofluorescence staining. Left, cell population was divided into five groups based on DNA stain and EdU incorporation, as indicated on the coloured scatter plots. G1: EdU negative and G1 DNA content, Early S: EdU positive and G1 DNA content, Mid S: EdU positive and intermediate DNA content, Late S: EdU positive and G2 DNA content. Significance was determined by Mann–Whitney, mean and S.D. of representative experiment of three biological repeats are shown. Right, total EdU intensity from $n = 3$ experiments normalised to siCont value at each part of S phase, significance determined by student's $t$-test. **b** RPE1 cells were pulse labelled with EdU for 30 min 24 h following transfection and flow cytometry used to quantify PCNA and MCM7 content on chromatin, flow cytometry plots shown from one of $n = 3$ experiments, values are normalised to G1, mean and S.D. are shown. Cell population was divided based on DAPI (DNA content) and EdU staining as in (**a**). Significance determined by 2-way ANOVA with Dunnett's multiple comparisons test. **c** RPE1 cells were analysed by DNA fibre analysis 24 h following transfection with the indicated siRNA. Mean and S.D. of fibre lengths shown from $n = 3$ experiments, Mann–Whitney test used to determine significance, at least 150 fibres were counted per condition. Scale bar represents 10 μm.

events have a minimal impact on overall DNA synthesis rates ensuring that S phase is completed in a timely manner (Fig. 4f).

A large range of proteins involved in DNA replication depend on the E2F family of transcription factors for their expression[13,15,23]. Controlling the availability of these proteins is crucial for faithful DNA replication, with an individual increase or a decrease in many of these components shown to be a potential source of replication stress leading to decreased replication fork speed[23,24,40–46]. In some cases, the slowing down in replication fork speed can be rescued by the addition of a single

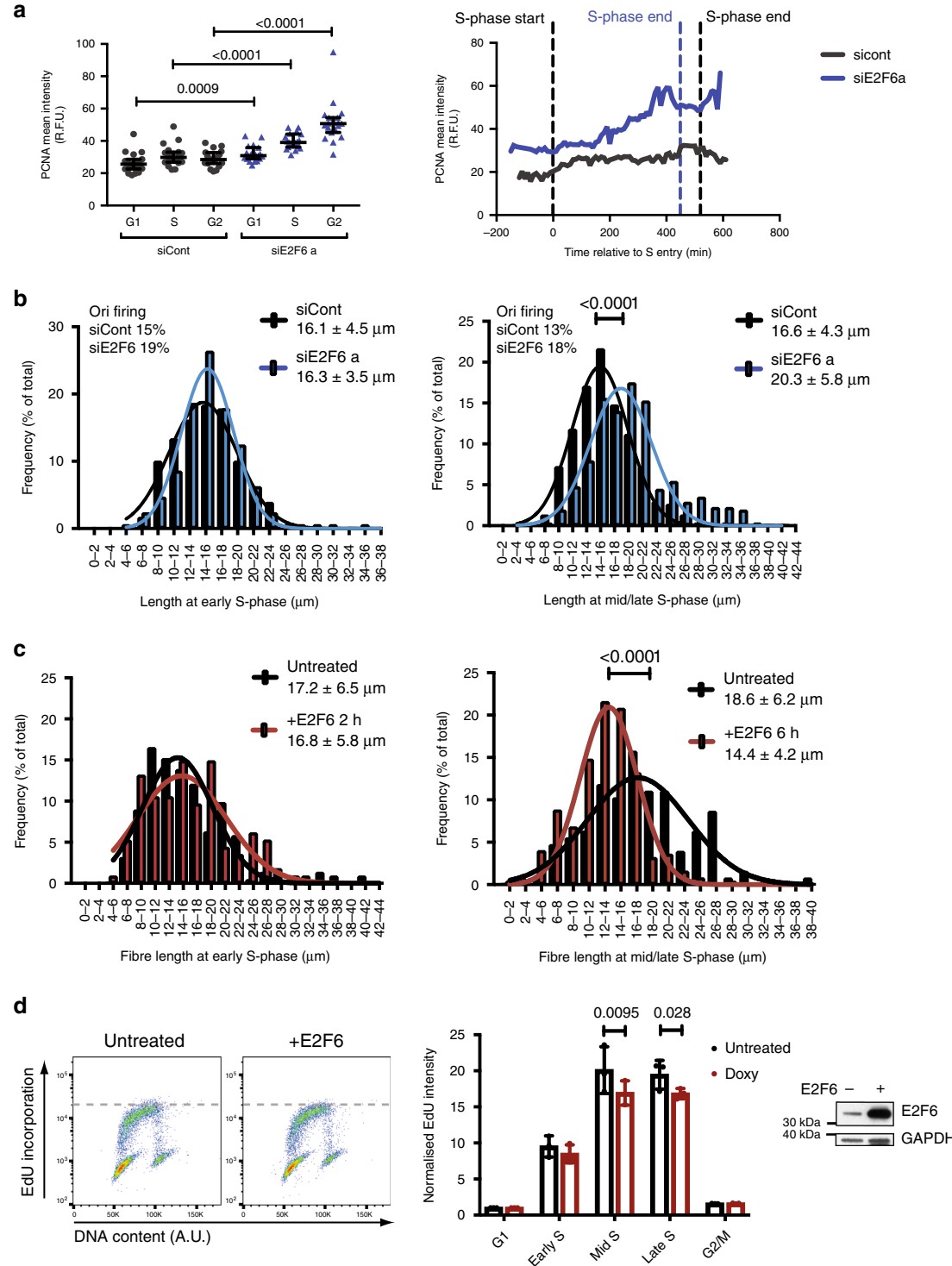

component, e.g. nucleosides[10,41]. However, nucleoside addition does not speed up forks in unperturbed conditions[47], suggesting that while maintaining fork speed depends on individual factors it is unlikely to be limited by the availability of one single factor. We propose that the availability of many replication components together, regulated by the E2F family of transcription factors, determine the replication capacity of a cell, which, depending on the number of active replication forks, limits fork speed.

This is consistent with the observations that an increase in replication fork speed can be achieved with a concurrent decrease

in active replication forks[5,8,10] and our data show that an increase in E2F activity allows for increased fork speed. This 'speeding up' of S phase contributes to genomic instability and cell-cycle arrest, which is in line with recently published data that shows that increasing fork speed above a certain point causes genomic instability[11]. This suggests that limiting replication capacity, and thereby fork speed, is important to maintain genomic stability. Taken together, replication capacity control through E2F-dependent transcription presents a robust mechanism to provide plasticity to changes in replication dynamics during S phase

**Fig. 3 Modulation of E2F-dependent transcription alters replication dynamics during S phase. a** RPE1 hTERT cells with endogenously tagged p21–EGFP and mRuby-PCNA were reverse transfected and imaged for 56 h, cells in the second cell cycle after transfection were scored. Mean intensity of mRuby-PCNA was calculated in single nuclei in G1, S and G2 phases. Twenty cells per condition were scored in $n = 2$ experimental repeats, median and interquartile range shown. Right, representative trace of the PCNA-mRuby mean intensity is shown (S phase entry at 0 min) for siCont (S phase end 540 min) and siE2F6a (S phase end 430 min). **b** RPE1 hTERT cells were synchronised by contact inhibition and transfected upon release. Cells were analysed by DNA fibre analysis at early (21 h) and mid/late (24 h) S phase of the first cell cycle following release. One histogram shown for each timepoint with mean, S.D. and origin firing frequency, at least 200 fibres were counted per condition. Representative experiment of $n = 3$. The percentage of origin firing during the first labelling time is shown. **c** RPE1 TetON E2F6 cells were treated as in (**b**), 4 μg/ml doxycycline was added 18 h after release to induce E2F6 overexpression, and cells were collected for DNA fibre analysis at 20 h and 24 h following release. Representative histograms shown for each timepoint with mean and S.D. from three independent experiments, significance was determined by the non-parametric Mann–Whitney test, at least 200 fibres counted per condition. **d** Cells were treated as in (**b**) and pulse labelled with EdU for 30 min at a mid-S phase timepoint (21 h), statistical significance was determined using a two-way ANOVA, normalised to G1, $n = 3$, mean and S.D. shown. Asynchronous RPE1 TetON E2F6 cells were treated with 4 μg/ml doxycycline 6 h before collection of whole cell extract by western blot, GAPDH is used as loading control (specific band indicated by star). Significance was determined by two-tailed Mann–Whitney test (**a–c**), 2-way ANOVA with Dunnett's multiple comparisons test within each cell cycle group (**d**).

to keep fork speed within an optimal range while maintaining DNA synthesis rates and timely completion of genome duplication (Fig. 4e).

While during an unperturbed cell-cycle E2F6 limits the replication capacity in mid and late S phase, in response to replication stress E2F6 is inactivated to allow E2F-dependent expression to maintain the levels of a large range of proteins involved in DNA replication, which allows resumption of replication[20,48]. Oncogene-induced replication stress has been identified as an important driver of cancer initiation. Deregulation of replication-initiation events is an important cause of oncogene-induced replication stress and is thought to be closely linked to the mis-regulation of E2F-dependent transcription in cancer cells[13,23,25,45,46,49,50]. Our data show that in addition to this the levels of E2F-dependent transcription in S phase also affects the replication potential and thereby the amount of DNA a cell can synthesise per unit time. Understanding how oncogene activity deregulates E2F transcription and the effect of these perturbations on DNA replication-initiation events and replication potential will be vital to the understanding of cancer biology. Overall, our work suggests that the tight regulation of E2F transcription in S phase is an important determinant in DNA replication control. Further research is required to establish a potential role for this regulation in maintaining genomic stability, which would have important implications for basic biology and the understanding of cancer.

## Methods

**Cell culture, drugs and siRNA transfection.** T98G, RPE1 hTERT, RPE1 TetON E2F6[48] and RPE1 mRuby-PCNA p21-GFP cells[51] were from ATCC and maintained in DMEM or DMEM/F12 (Gibco) supplemented with 10% FBS (Sigma), sodium bicarbonate (Gibco) and penicillin/streptomycin (Gibco). RPE1 TetON E2F6 cells were maintained in 5 μg/ml Blasticidin and 100 μg/ml Zeocin. Doxycycline was used at 4 μg/ml, nocodazole at 100 ng/ml and roscovitine at 25 μM. Non-targeting (referred to as siCont unless otherwise stated) and E2F6 pooled siRNAs (referred to as siE2F6a) were purchased from Dharmacon (ON-TAR-GETplus E2F6 siRNA L-003264-00-0005). Other siRNA sequences used were LacZ: AACGUACGCGGAAUACUUCGA, siE2F6 b: AAACAAGGUUGCAACGAAA UU. Lipofectamine RNAiMAX transfection reagent (Invitrogen, 13778-075) and OptiMEM (Gibco) were used for siRNA transfection according to manufacturer's instructions.

**Immunofluorescence.** Cells were pre-extracted for 1 min in ice cold PBS 0.2% Triton-X100, fixed with 4% formaldehyde for 20 min. Coverslips were blocked in 1% BSA for 1 h and incubated in primary antibodies overnight at 4 °C, RPA32 (RPA2) (Ms, MABE285, 1:500) and Phospho-Histone H2A.X (γH2AX) (Ser139) (Rb, 20E3, Cell Signalling Technology, 1:400). Coverslips were incubated in secondary antibodies for 1 h at RT; anti-mouse Alexa Fluor 488 and anti-rabbit Alexa Fluor 647 1:2000 (Life Technologies). Coverslips were then incubated in Hoechst (Invitrogen) 1:10,000 for 5 min and mounted with Fluoroshield (Sigma). Images were obtained by confocal microscopy with a Leica TCS SP5 or SPE2 ×63 objective lens using LASAF. Images were processed in Fiji.

For EdU incorporation evaluated by microscopy cells were reverse transfected and seeded at density 6000 cells/well in 96-well plate (PerkinElmer, CellCarrier).

Twenty-four hours after transfection, cells were incubated with 10 μM EdU for 30 min, then were fixed with 4% formaldehyde for 20 min, permeabilised with 0.5% TritonX in PBS for 5 min and EdU was detected by ClickIT reaction (Life Technologies). Cells were stained with 2 μg/ml DAPI solution in PBS for 30 min. Images were acquired by Opera Phenix HCS System with Harmony (PerkinElmer) and analysed in Columbus software (PerkinElmer) and custom-made RStudio script available upon request.

**DNA fibre analysis.** Fibre labelling and spreading was performed as in ref. [31], RPE1 hTERT or RPE1 TetON E2F6 cells were incubated for 20 min with 20 μM CldU then 250 μM IdU. Cells were trypsinised and resuspended in PBS before spreading in buffer (200 nM Tris pH 7.4, 50 nM EDTA, 0.5% SDS). Slides were incubated in MeOH/AcOH (3:1) for 10 min at room temperature before immunostaining. Images were taken by confocal microscopy and analysed with Fiji. 150–200 fibres were measured per condition for each experiment. Curves on histograms represent Gaussian fits of the data. The origin firing was scored considering just the origins starting in the first labelling time (green-red-green) relative to total (ongoing + origins).

**Flow cytometry.** Flow cytometry was performed for DAPI/PCNA/MCM2/MCM7/EdU as in ref. [29]. Cells were incubated with 10 μM EdU for 30 min before trypsinisation and collection. Cells were pre-extracted in CSK buffer for 10 min on ice before fixation in 4% formaldehyde for 10 min at RT. Cells were permeabilised in ice cold 70% ethanol before incubation for 1 h at RT in primary antibody followed by secondary antibody. EdU was then detected using the Click iT-EdU assay (C10634 Life Technologies) according to the manufacturer's instructions and cells were treated with DAPI (0.5 μg/ml). PCNA PC10 sc-56, MCM2 sc9839, MCM7 sc56324 from Santa Cruz and MCM3 A300-192A from Bethyl Laboratories, anti-mouse Alexa Fluor 488 from Life Technologies. Samples were measured on a BD-LSR II flow cytometer using DIVA software (BD) and analysed using FlowJo software.

**Live-cell imaging and analysis.** RPE1-hTert cells with endogenously tagged mRuby-PCNA and p21–EGFP were used[27,51]. 1000 cells/well were plated in 384-well CellCarrier (PerkinElmer), cells were reverse transfected, and imaging began 8 h later. Live-cell imaging was performed on the Opera HC spinning disk confocal microscope (PerkinElmer), with atmospheric control to maintain cells at 37 °C, 5% CO₂ and 80% humidity. Cells were imaged using a ×20 (N.A. 0.45) objective at 10 min intervals for 72 h in Phenol-Red Free DMEM supplemented with 10% FBS and 1% P/S.

Image processing was performed using NucliTrack software[28,51], cell-cycle phase transitions were evaluated as described in ref. [26] based on PCNA signal pattern changes.

First generation (mother cells) are defined as those able to finish mitosis before 30 h following transfection. Second generation (daughter) and third generation cells (grand-daughter cells) are cells arising from the first and second generation, respectively.

Intensity of p21 and PCNA for each cell in every timepoint was measured by NucliTrack software.

The data obtained from NucliTrack were further processed with custom-made scripts in RStudio and Perl to count p21 and PCNA intensities per each phase of the cell cycle and generate graphs of PCNA traces. The software codes are available upon request.

**Western blot.** Samples were prepared in RIPA buffer (Tris-HCl pH 7.5 20 mM, NaCl 150 mM, EDTA 1 mM, EGTA 1 mM, NP40 1%, NaDoc 1%) containing phosphatase inhibitor cocktails 2 and 3 1:1000 (Sigma P5736 and P0044) and protease inhibitor cocktail 1:1000 (Sigma, P8430). Whole-cell lysates were loaded onto 4–12% Bis-Tris Novex precast gels (Thermo Fisher). Antibodies were used at 1:1000 unless otherwise stated: Cdc6 (Ms, sc9964, 1:500), Cdc7 (Ms, sc56274),

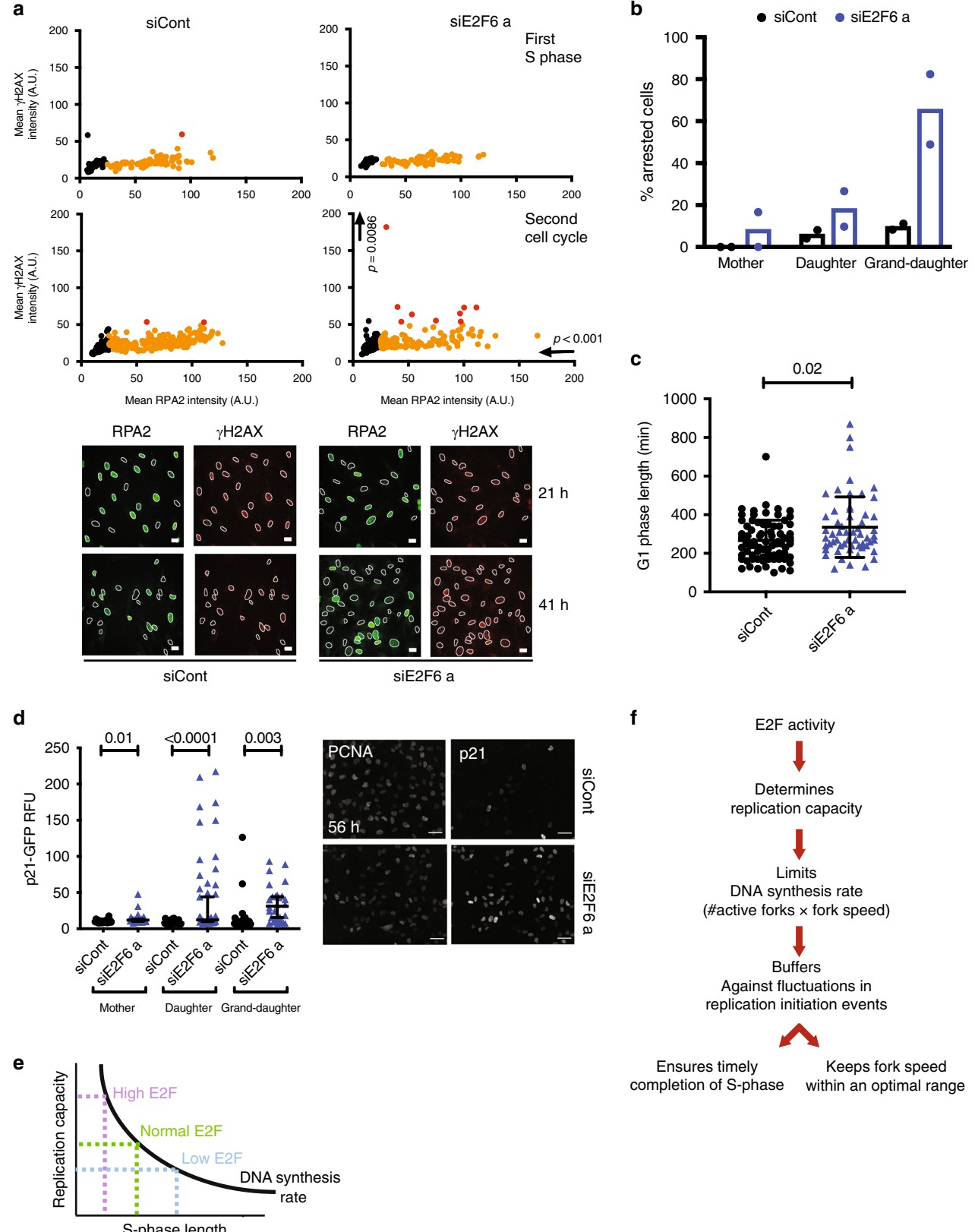

Cyclin A (Ms, sc53227), PCNA (Ms, sc56291) from Santa Cruz, GAPDH (Ms, GT239, 1:3300) from GeneTex, E2F6 (Rb, ab155978) Abcam.

**Chromatin preparation**. Buffer A (Hepes, pH 7.9, 10 mM, KCl 10 mM, MgCl 1.5 mM, sucrose 0.34 M, glycerol 10%, DTT 1 mM and protease and phosphatase inhibitors as recommended by the supplier). Buffer B (3 mM EDTA, 0.2 mM EGTA and protease and phosphatase inhibitors). The nuclear fraction was pelleted at $1300 \times g$, 5

min, 4 °C. The chromatin fraction was pelleted at $1700 \times g$, 5 min, 4 °C. The sample was spun at $9600 \times g$, 5 min, 4 °C before use. Alternatively, the Abcam Chromatin kit (ab117152) was used according to manufacturer instructions.

**Statistics**. Graphpad Prism software was used for statistical tests. Figures 1f, g, 4c and Supplementary Figs. 1d, e, 4c statistical significance was calculated against control conditions using the non-parametric Kruskal–Wallis test with Dunn's

**Fig. 4 An increase in E2F-dependent transcription in S phase results in accumulating DNA damage in later cycles. a** RPE1 cells were synchronised by contact inhibition, reverse transfected upon release into the cell cycle, and levels of chromatin-bound RPA2 and γH2AX were quantified by immunofluorescence during the first S phase (21 h) and second cell cycle (41 h) after release. Black, non-S phase cells (RPA2 < 50 A.U.); orange and red dots, low and high levels of γH2AX, respectively (arbitrary γH2AX threshold = 50 A.U.). Data are representative of three independent experiments, scale bar represents 20 μm. **b** Percentage of arrested and dividing cells were quantified from live imaging of RPE1 mRuby-PCNA p21-GFP cells through two divisions. Mean and S.D. from two independent experiments shown. **c** Length of G1 phase calculated from live imaging of RPE1 mRuby-PCNA p21-GFP. siCont $n = 75$, siE2F6, $n = 58$. Mean and S.D. are shown. **d** Quantification of p21 levels in RPE1 mRuby-PCNA p21-GFP cells through two divisions. Images of p21 and PCNA fluorescence intensity 56 h after commencement of imaging, scale bar = 50 μm. Median with 95% confidence intervals are shown. siCont $n = 22$, 22 and 25 mother, daughter and grand-daughter cells, respectively, siE2F6a $n = 33$, 30, 44. **e, f** E2F activity determines replication capacity, which limits DNA synthesis rate to provides a general mechanism to ensure timely completion of S phase largely independent of replication-initiation events. Significance was determined by a two-tailed Mann–Whitney test for each marker compared to control at each timepoint (**a**) and compared to control within each generation (**d**), and Kruskal–Wallis test with Dunn's multiple comparisons test (**c**).

multiple comparisons test. Figures 2b, 3d and Supplementary Fig. 2c, f significance was determined using a two-way ANOVA with Dunnett's multiple comparisons test and with Sidak's multiple comparisons test in Supplementary Figs. 2c, f and 4f. For DNA fibre analysis in Figs. 2c, 3b, c and Supplementary Figs. 2g, h and 3c, d significance was determined between distributions of measurements using the non-parametric two-tailed Mann–Whitney test. At least 150 fibres were measured per condition. Figure 2a, right, Supplementary Figs. 3e, f and 4b significance was determined by a two-tailed student's t-test. Figures 2a (left), 3a, 4a, b, d and Supplementary Figs. 2a and 4a, e significance was determined using the non-parametric two-tailed Mann–Whitney test. Supplementary Figure 1a, significance was determined with a one-way ANOVA.

**Reporting summary**. Further information on research design is available in the Nature Research Reporting Summary linked to this article.

## Data availability

The source data underlying Figs. 2a, b and 3d, Supplementary Figs. 1b, 2c–f, 3a, c and 4f are provided as Source data files. All datasets and scripts generated and/or analysed during the current study are available from the corresponding author upon reasonable request. Source data are provided with this paper.

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

## Acknowledgements

This work was supported by core funding to the MRC-UCL University Unit (Ref MC_U12266B) and funded by R.d.B.'s Cancer Research UK Programme Foundation Award CRUK CDF: C63833/A25729. A.R.B. is funded by CRUK CDF: C63833/A25729 and MRC core funding to the London Institute of Medical Sciences.

## Author contributions

B.P., C.B. and R.d.B. designed research; B.P., S.P., A.B. and C.B. performed research; B.P., E.V., T.S. and C.B. analysed data; and B.P., C.B. and R.d.B. wrote the paper.

## Competing interests

The authors declare no competing interests.
