## [Peer Review File · Nature Communications]

Reviewers' comments:

Reviewer #1 (Remarks to the Author):

In this manuscript, Pennycook et al. intend to propose a new concept, called 'replication capacity', which the authors define as the maximum amount of DNA a cell is able to synthesize per unit of time throughout S phase. The authors argue that E2F-dependent transcription determines the replication capacity of a cell. The manuscript is well written, however, the conclusions are overstated and the experimental evidence is rather preliminary. Some of the results are inconsistent between experiments and some statistical differences are arguable. In general, the authors test the effect of knocking-down E2F6 during S phase. With this, the authors claim in the title that E2F activity determines replication capacity, which is an overstatement. The comments below indicate the multiple problems with this manuscript, in terms of experimental design, controls and interpretation. Overall, the present results do not support the author's model and conclusions. I suggest addressing at least some of the specific comments listed below before submitting the manuscript elsewhere.

Major comments:

1. The authors modulate E2F activity only via E2F6, but claim that general E2F activity is the driver for replication capacity (the maximal amount of DNA a cell is able to synthesise per unit time throughout S phase). Proof for their hypothesis falls short as no general E2F activity readout is provided and a positive E2F transcription factor (e.g. E2F1) is not modulated whatsoever. The authors should therefore focus on claims for E2F6-dependent mechanisms on the title and their conclusions or include more comprehensive E2F readouts.
2. Related to 1 above, the authors should also evaluate the effect of knocking-down E2F4 and E2F3 (Giangrande et al., *Genes and Dev*, 2004) in S phase using RPE1 as a cellular model. (currently only E2F6 has been looked at).
3. Many controls are missing in the experimental settings of the cell cycle profile analysis. Cells were synchronized by serum starvation and then treated with nocodazole. The authors should therefore include siControl and siE2F6 cell cycle profiles without synchronization, only synchronized without nocodazole treatment and follow them up to 48 h.
4. The assessment of DNA damage (Fig. 4) is not sufficient. No proper readouts of replication stress, such as RPA foci detection or 53BP1 body formation in the following G1 were used. For RPA2 foci detection, cells have to be pre-extracted before fixation and analysis, as free nucleoplasmic RPA would largely abolish foci detection. Alternatively, WB analysis for phosphorylated ATR or Chk1 could have been used. The mere fact that higher levels of p21 exist in E2F6-depleted cells in the second cell cycle after G1 release shows that replication stress must be present in the first cell cycle. The provided analysis is not sufficient and comprehensive enough to draw any conclusions.
5. Figure 1b. The authors should show western blots of some representative examples of cyclins, licensing factors, E2F6a, DNA synthesis and repair proteins and repressors of G1/S, by synchronizing cells as in C and blotting different proteins after the release.
6. Figure 1c. These cell cycle profiles of T98G cells should be replaced by an equivalent experiment using RPE1 cells. T98G cells are p53 mutant and clearly progress differently through the cell cycle after arrest than RPE1 cells (see S3b). Especially that this is the only one experiment where the authors used T98G. Perhaps it is due to some historical reasons that this cell line is included, as the authors have published other papers using T98G (Bertoli et al., *Curr Biol*, 2013).

7. Figure 2a. A color code should be presented in the scattering plots for siControl and siE2F6, including gating strategy. The differences in EdU intensity are not convincing and the results from triplicates should be presented as a bar plot with S.D.

8. Figure 2b. The authors must measure the distance between origins using DNA fibre/combing techniques. PCNA intensity is not a direct measure of origin activity. PCNA is also needed to repair DNA and it can be engaged in replication factories for longer time in the presence of arrested forks. In any case, the authors should present the scattering plots from flow cytometry from siControl and siE2F6 cells. In the current figure, it is not clear, which cells were used as the example.

9. Figure 2c. Approximately 200 forks were counted for each condition. Perhaps the number of analyzed forks is not sufficient, as there is a discrepancy in the fork length with other similar experiments. There is plenty of room in the supplementary figures to show individual fibre experiments to judge the difference between experiments.

10. 'E2F-dependant transcription in S phase results in an increase in replication fork speed, thus increasing the overall DNA synthesis rate without any detectable change in origin licensing or firing' – again this sentence is overstated. The authors should confirm their findings using classical biochemistry experiments, like tritium-labelled thymidine incorporation and measurement of the ori-to-ori distance by immunofluorescence.

11. Why do siE2F6a-treated cells show no faster S phase after release from contact inhibition (Suppl. Fig 3), but faster replication forks? This contradicts the main message of the paper. It rather seems that contact inhibition and serum starvation have different impact on S phase length after E2F6-depletion despite increased fork speed, which is not addressed by the authors (Fig. 1 and Suppl Fig. 1 in contrast to Suppl. Fig. 3).

This study is indicative:

Yang HW, Chung M, Kudo T, Meyer T. Competing memories of mitogen and p53 signalling control cell-cycle entry. *Nature*. 2017 Sep 21;549(7672):404-408. doi: 10.1038/nature23880. Epub 2017 Sep 6. PubMed PMID: 28869970.

12. Also, the cell cycle profiles in the Figure S3b at 24 h look similar between siControl and siE2F6. The authors should quantify the amount of cells in early-mid-late S by EdU incorporation after the release

13. Figure 3a. The authors should confirm their findings by investigating the level of PCNA by immunoblotting in the same experimental settings.

14. Figure 3b. Interestingly, the authors found that the fork length was similar in siControl cells in early and mid/late S phase, (the average length in these cells was ~16 μ m). In the Figure 2c, for siControl the average length in non-synchronized cells was 17.8. Is this difference due to technical problems of the DNA fibres or due to synchronization? Fibre results from equivalent experiments should be presented and the number of scored fibres for each experiment should be increased.

15. Figure 3c. Is the fork length significantly different between early vs mid/late S, 17.2 vs 18.6? Any explanation why are these numbers different from the ones presented in the Figure 3b for siControl untreated cells?

16. Figure 4. γ H2AX and RPA intensities can be correlated with DNA content. The cell cycle profiles from generations 0, 1, 2 should be shown. I speculate that the cells start to arrest in G1 due to DNA damage.

Minor comments:

- It probably is just semantic but the conclusion from results presented in the Figure 1 are overstated (Page 4, the end of paragraph 1). A more precise conclusion should be – kd-E2F6 cells spend less time in S phase.
- In line 106, incomplete sentence: “This allowed us to monitor.”

Reviewer #2 (Remarks to the Author):

The study by Pennycook et al. analyzes the role of E2F in controlling S phase length by determining what the authors term “replication capacity”. Replication capacity is a novel and useful parameter of a cell’s DNA synthesis kinetics. The authors essentially demonstrate that E2F regulates the expression of S phase factors and actively adjusts those expression levels to ensure completion of genome duplication with a given number of active replication origins. Importantly, the study modulates E2F6, which represses E2F-driven transcription of S phase proteins in late S phase. The authors show convincingly that E2F6 knockdown accelerates S phase progression in 2 different (a transformed and nontransformed) cell lines by increasing fork speed in late S phase. The data is well-controlled, statistics are solid and the conclusions are well justified. Previous studies have shown that an increase in fork speed is associated with DNA damage. The authors show here that this is not necessarily a direct effect. This is an important observation. The finding that low level damage can be observed in the second cell-cycle after E2F6 knockdown is theoretically a consequence of diminished origin licensing, which would explain why p21 expression is increased and G1 is extended in the KD cells. This has not been explored rigorously but could be easily done by analyzing Mcm2 and Mcm7 chromatin association at the end of G1 of the second cycle. Examining this point would make this particular section of the paper clearer.

Overall, this is a rigorous and important study.

Reviewer #3 (Remarks to the Author):

In this manuscript Pennycook et al. explore the impact of E2F transcriptional activity de-repression in S-phase dynamics of human cells. Authors observe that increasing E2F transcription by interference of the repressor E2F6 results in a faster S-phase completion. A slight increase in EdU incorporation is observed in E2F6 interfered cells, which correlates with longer replication tracks in DNA fibers, suggesting that enhanced E2F transcription increases fork speed. Conversely, overexpression of E2F6 in S-phase reduces replicated fiber length and EdU incorporation. Lastly, a slight increase is observed in γH2AX phosphorylation and RPA2 intensity, proxies of DNA damage signaling and replication stress, respectively, in E2F6 interfered cells, which correlate with an increase in p21 foci and slight lengthening in G1 duration. Based on these results authors propose that E2F controls the replication capacity of the cell (defined by fork speed and number of active forks), as part of a mechanism that would respond to fluctuations in replication initiation events to maintain absolute synthesis rates and S-phase duration.

The data presented are in general of good quality and the conclusions are original and of potential interest for researchers in the replication and cell cycle fields. However, there are some issues about experimental design and data interpretation raising concerns that should be addressed to support the main conclusions and for the model to be convincing.

Major points:

- Authors define the concept of a replication capacity, which should be kept somewhat constant to promote optimal S-phase length and replication speed, and propose that E2F transcription works within a mechanism controlling replication capacity. Replication rates and fork speed seem increased upon E2F6 interference. However, to conclude that replication capacity is enhanced by E2F transcription in this context authors should prove that the frequency of initiation events is unchanged. This should be straight forward, as origin firing frequencies could be measured in the double label fiber experiments shown. MCM7 data on figure 2b seem to be an estimate of nuclear MCM levels rather than a measure of origin licensing/firing.
- Authors should provide evidence that E2F transcription responds to changes in replication initiation event numbers to support the notion that it acts within a mechanism regulating replication capacity.
- A key point is which E2F targets promote an enhancement of replication fork speed. A straightforward mechanism would be upregulating dNTP pools via induction of ribonucleotide reductase expression. Authors should measure dNTP levels in control and E2F6 interfered cells to rule out this possibility.

Minor point:

- DNA damage signaling and replication stress are addressed 21 hours after cell cycle block release when a small proportion of cells (5-7%) is in S-phase. These experiments would be more informative if performed at later time points when more cells have ongoing replication.

We would like to thank the reviewers for their helpful comments and suggestions. Please find below our point by point rebuttal addressing their comments with changes to the text indicated in bold in the text of the revised manuscript.

Reviewer #1 (Remarks to the Author):

In this manuscript, Pennycook et al. intend to propose a new concept, called 'replication capacity', which the authors define as the maximum amount of DNA a cell is able to synthesize per unit of time throughout S phase. The authors argue that E2F-dependent transcription determines the replication capacity of a cell. The manuscript is well written, however, the conclusions are overstated and the experimental evidence is rather preliminary. Some of the results are inconsistent between experiments and some statistical differences are arguable. In general, the authors test the effect of knocking-down E2F6 during S phase. With this, the authors claim in the title that E2F activity determines replication capacity, which is an overstatement. The comments below indicate the multiple problems with this manuscript, in terms of experimental design, controls and interpretation. Overall, the present results do not support the author's model and conclusions. I suggest addressing at least some of the specific comments listed below before submitting the manuscript elsewhere.

We would like to thank reviewer 1 for pointing out sections that need further clarification to highlight the generality of our findings. We have been able to address all concerns raised by the reviewer by including new data and highlighting already included and previously published data. Overall, reviewer 1's main concern seems to be the generality of our findings based on the argument that we do not present any data showing that loss of E2F6 increases E2F-dependent transcription during S phase. In addition, they suggest that alternative ways of increasing E2F-dependent transcription during S phase are needed to proof our hypothesis or else we should specify that this is a E2F6 specific phenomena. However, our previously published work (Bertoli et al Current Biology 2013) clearly establishes that knocking down E2F6 has a general effect on E2F activity (clarified in more detail below). Importantly, this work shows that knocking down E2F6 only affects E2F-dependent transcription during S phase, which is essential for our current study, which focusses on the role of E2F activity during S phase, which we have further specified in the abstract.

Major comments:

1. The authors modulate E2F activity only via E2F6, but claim that general E2F activity is the driver for replication capacity (the maximal amount of DNA a cell is able to synthesise per unit time throughout S phase). Proof for their hypothesis falls short as no general E2F activity readout is provided and a positive E2F transcription factor (e.g. E2F1) is not modulated whatsoever. The authors should therefore focus on claims for E2F6-dependent mechanisms on the title and their

conclusions or include more comprehensive E2F readouts.

Our previously published work (Bertoli et al Current Biology 2013) establishes, using microarray, that knocking down E2F6 increases general E2F activity during S phase. In that paper the microarray data is further confirmed by the analysis of a large panel of well-known E2F targets via Reverse Transcriptase quantitative PCR, which include licensing and replication factors, cell cycle regulators and activator E2Fs, relevant to the current study. Overall, we believe that our published results, and that of others, are sufficient proof of the notion that depleting E2F6 increases E2F-dependent transcription in S phase.

In addition to these published results we present additional data in the current study (Supplementary Figure 3a) showing that the levels of many E2F target proteins, relevant for this study, such as MCM2, CtIP, Cyclin E, Cyclin A, Cdc45, PCNA, increase upon E2F6 depletion, and decrease upon E2F6 overexpression, as expected.

2. Related to 1 above, the authors should also evaluate the effect of knocking-down E2F4 and E2F3 (Giangrande et al., Genes and Dev, 2004) in S phase using RPE1 as a cellular model. (currently only E2F6 has been looked at).

The current study focuses on increasing E2F-dependent transcription during S phase. We agree that there are alternative ways of increasing E2F-dependent transcription, but argue that these either do not increase activity during S phase (examples of this are E2F3 and E2F4, as suggested by the reviewer, discussed in more detail below) or do not confine an increase only during S phase and are therefore not suited for the current study.

The Giangrande et al., study, the reviewer refers to, shows that knocking down E2F4 has no, or little, effect on E2F-dependent transcription during S phase. It shows that only in the case of chronic loss of E2F6 (MEFs deriving from E2F6 knock out mice) is E2F4 able to bind E2F-target promoters during S phase to compensate for the loss of E2F6. Only in this scenario, in cells lacking E2F6, does depletion of E2F4 increase E2F activity in S phase. Our published work and that of others show that acute depletion of E2F6 increases E2F-dependent transcription during S phase, indicating that E2F4 does not compensate for E2F6's function in this context.

In addition, it is well established that E2F4 and E2F3b (E2F3a is an activator) requires pocket proteins (p107 and/or p130) for its general repressor activity. Cyclin-CDK activity during S phase compromises the ability of these pocket proteins (which are inactivated by CDK-dependent phosphorylation), and therefore E2F4 and E2F3b's potential, to repress transcription during S phase.

Overall, a large body of evidence suggests that E2F4 has an important role in repressing E2F-dependent transcription when cells return into interphase and

exit the cell cycle in G1, whereas E2F3b has been suggested to regulate a specific subset of genes required to promote differentiation (Asp, P., et al., G&D 2009), likely during G1.

3. Many controls are missing in the experimental settings of the cell cycle profile analysis. Cells were synchronized by serum starvation and then treated with nocodazole. The authors should therefore include siControl and siE2F6 cell cycle profiles without synchronization, only synchronized without nocodazole treatment and follow them up to 48 h.

We initially performed the cell cycle profile analysis in the absence of nocodazole (data shown below). However, whilst the data does show accelerated S phase completion, the ability of cells to return to interphase in this setting masks the timely completion of S phase during the first cell cycle. We do not observe a significant difference in cell cycle distribution in non-synchronized cell populations during 48 h of culturing.

Figure 1 T98G cells were synchronised by serum starvation for 72h following transfection with the indicated siRNAs, and were re-transfected before release into the cell cycle. Samples were collected for FACS analysis of DNA content at the indicated times following release.

4. The assessment of DNA damage (Fig. 4) is not sufficient. No proper readouts of replication stress, such as RPA foci detection or 53BP1 body formation in the following G1 were used. For RPA2 foci detection, cells have to be pre-extracted before fixation and analysis, as free nucleoplasmic RPA would largely abolish foci detection. Alternatively, WB analysis for phosphorylated ATR or Chk1 could have been used. The mere fact that higher levels of p21 exist in E2F6-depleted cells in the second cell cycle after G1 release shows that replication stress must be present in the first cell cycle. The provided analysis is not sufficient and comprehensive enough to draw any conclusions.

We apologize to the reviewer for not making it clearer in the text that the experiments shown in Figure 4 do indeed assess both DNA damage (as assessed by chromatin bound γ H2AX signal indicative of ATM activity) and replication stress (as assessed by the chromatin bound fraction of RPA, the ssDNA binding protein). We have amended two sentences to clarify this in the

text:

“We therefore tested the levels of DNA damage checkpoint activation, **via γ H2AX intensity on chromatin**, upon E2F6 depletion in synchronized cells.”

“However, we did observe a small but significant increase in γ H2AX chromatin staining **and in the number of cells showing at least one γ H2AX focus** in siE2F6 treated cells compared to control cells during the second cell cycle (41h)(Fig. 4a, Supplementary Fig. 4b).”

Importantly the dot-plots show the total intensity of γ H2AX and RPA signal in single nuclei in the first and second cell cycles following transfection. Correlating both γ H2AX (DNA damage) and RPA (replication and replication stress) signal at the single cell/nuclei level allows for monitoring replication stress-induced DNA damage levels and is widely used in the field. The interpretation of the data shown in Figure 4a is that cells with higher RPA signal (shown in orange) are cells in S phase, the ones with low RPA signal (black) are mostly in G1 phase and some in G2. The cloud of orange cells with high levels of γ H2AX signal (red) are cells in S phase experiencing DNA damage.

In addition to this, in the revised version of the manuscript we have added new data, quantifying γ H2AX foci in the second cell cycle (Supplementary Fig 4b), and the percentage of cells arrested in G1 upon E2F6 depletion (Figure 4b), both of which show an increase in siE2F6 treated cells.

5. Figure 1b. The authors should show western blots of some representative examples of cyclins, licensing factors, E2F6a, DNA synthesis and repair proteins and repressors of G1/S, by synchronizing cells as in C and blotting different proteins after the release.

As mentioned above, this data has been presented in a previous publication (Bertoli et al. 2013 Current Biology), but we have now included similar data in Supplementary figure 3a for the RPE1 cells used in this study.

6. Figure 1c. These cell cycle profiles of T98G cells should be replaced by an equivalent experiment using RPE1 cells. T98G cells are p53 mutant and clearly progress differently through the cell cycle after arrest than RPE1 cells (see S3b). Especially that this is the only one experiment where the authors used T98G. Perhaps it is due to some historical reasons that this cell line is included, as the authors have published other papers using T98G (Bertoli et al., Curr Biol, 2013).

T98G cells have been used widely to study E2Fs and are well characterized. Importantly T98G cells arrest and release synchronously by serum starvation, which is the main reason they were used in the initial experiment to assess cell cycle progression. We agree with the reviewer that since T98G cells are transformed there are issues such as the mentioned p53 mutation, and also the

content of DNA being higher than diploid. This is why we have limited their use in the current study to just the initial cell cycle profile synchronization experiment.

In addition, synchronization of a cell culture, including serum starvation and contact inhibition, by its very nature perturbs cell cycle progression. The best way to measure cell cycle progression is therefore in unperturbed conditions via live imaging of single cells in a population using an endogenous tag to distinguish cell cycle phases. So rather than synchronizing RPE1 cells to confirm our data in T98G cells we used live imaging of RPE1 cells with endogenously tagged mRuby-PCNA and p21-GFP. These data confirm the observation made in T98G, and are in line with the main conclusions of the manuscript.

7. Figure 2a. A color code should be presented in the scattering plots for siControl and siE2F6, including gating strategy. The differences in EdU intensity are not convincing and the results from triplicates should be presented as a bar plot with S.D.

We have added colour coding to the representative plots. We have added normalised data from triplicate experiments in Fig 2a and have added the repeats of the experiment in Supplementary Fig 2a.

8. Figure 2b. The authors must measure the distance between origins using DNA fibre/combing techniques. PCNA intensity is not a direct measure of origin activity. PCNA is also needed to repair DNA and it can be engaged in replication factories for longer time in the presence of arrested forks. In any case, the authors should present the scattering plots from flow cytometry from siControl and siE2F6 cells. In the current figure, it is not clear, which cells were used as the example.

We have added representative examples of scatter plots of flow cytometry data showing chromatin-bound PCNA in both siCont and siE2F6 treated cells. In addition, we have added data showing levels of chromatin-bound PCNA and Cdc45 by western blot (Supplementary Fig 2d). Whilst PCNA can only be detected on chromatin when it is loaded onto DNA by replication factors (Mailand et al 2013) and is widely used as a marker of ongoing DNA replication and replication foci (Leonhardt et al. 2000, Ekholm-Reed et al. 2004, Forment and Jackson 2015), we agree with the reviewer that it is also bound to chromatin at arrested forks. Therefore, we have now included measurements of DNA replication origin firing following E2F6 knockdown using our DNA fibre analysis data (Supplementary Fig2e). This data shows there is no significant difference between siCont and siE2F6 treated cells, confirming the PCNA results. A sentence was added to the text:

“In agreement with this finding, the amount of origin firing observed in DNA fiber spreads did not change upon E2F6 depletion (Supplementary fig 2e).”

“This is not coupled with a change in origin firing (Supplementary Fig. 3d).”

9. Figure 2c. Approximately 200 forks were counted for each condition. Perhaps the number of analyzed forks is not sufficient, as there is a discrepancy in the fork length with other similar experiments. There is plenty of room in the supplementary figures to show individual fibre experiments to judge the difference between experiments.

These data presented represents three biological replicates combined. We have now added an additional set of experiments in supplementary figure 2g, which further confirm these findings. The distribution displays measurements of DNA fibers from three biological repeats, whilst the mean and standard deviation from the three repeats shows what type of variation can be expected. Variation in the length of DNA fibers, from one experiment to the next, is inherent to fibre analysis and is due both to biological and technical variability (Técher et al. 2013). Small differences in timing, temperature and the nature of the DNA fiber experiment add noise that accounts for the differences in the average length from one experiment to the next. Because of this it is common practice to compare DNA fiber data from a single experiment testing different conditions. We observe an increase in fibre length upon E2F6 depletion compared to its control, in all replicate experiments confirming the reproducibility of these results. In addition, we confirm this using a different siRNA targeted against E2F6 (Supplementary Fig. 2g, right panel).

10. 'E2F-dependant transcription in S phase results in an increase in replication fork speed, thus increasing the overall DNA synthesis rate without any detectable change in origin licensing or firing' – again this sentence is overstated. The authors should confirm their findings using classical biochemistry experiments, like tritium-labelled thymidine incorporation and measurement of the ori-to-ori distance by immunofluorescence.

The techniques used in this study to assess DNA synthesis and licensing and firing are those currently used in the field. For example, using EdU incorporation to analyse DNA synthesis rates, gives comparable results to previously used tritium incorporation (Gratzner 1982, Salic and Mitchison 2008), but provides single cell resolution allowing to determine DNA synthesis per cell. We used two previously reported techniques to measure DNA origin licensing and firing - western blot and flow cytometric analysis of chromatin-bound licensing and firing factors (Moreno et al. 2016, Ekholm-Reed et al. 2004). Measurement of inter-origin distance (ori-to-ori) has been used previously as a readout for replication origin activity. However, the limited distance over which a continuous DNA fibre can be spread creates a bias for the measurement of inter-origin distances within replication domains, as these are shorter so more likely to be detected by a DNA fibre assay. It is now well recognised in the field that this will only provide a measurement of local, and not global, inter origin distance. Therefore, we decided to focus on measuring new replication origin activation events in our

DNA fibre assays, which confirms that there is no reduction upon E2F6 depletion and this data has been added to the manuscript (Supplementary Figs 2 and 3) (Técher et al. 2013, Nieminuszczy et al. 2016).

11. Why do siE2F6a-treated cells show no faster S phase after release from contact inhibition (Suppl. Fig 3), but faster replication forks? This contradicts the main message of the paper. It rather seems that contact inhibition and serum starvation have different impact on S phase length after E2F6-depletion despite increased fork speed, which is not addressed by the authors (Fig. 1 and Suppl Fig. 1 in contrast to Suppl. Fig. 3).

This study is indicative:

Yang HW, Chung M, Kudo T, Meyer T. Competing memories of mitogen and p53 signalling control cell-cycle entry. *Nature*. 2017 Sep 21;549(7672):404-408. doi: 10.1038/nature23880. Epub 2017 Sep 6. PubMed PMID: 28869970.

We agree with the reviewer that the effect on S phase length, seen in the single cell time lapse analysis, is not obvious in a synchronized RPE1 cell population. We believe that this is due to the lack of a robust synchronous release with RPE1 cells, which probably masks the decrease in S phase length in this setup. RPE1 cells are hard to arrest and synchronously release. Whilst the enrichment in the S phase population does allow us to observe an increase in DNA fibre length in middle/late S phase the extent of the synchrony is insufficient to evaluate S phase length. It is therefore, unfortunately, not possible to draw any conclusions about S phase length from this data.

12. Also, the cell cycle profiles in the Figure S3b at 24 h look similar between siControl and siE2F6. The authors should quantify the amount of cells in early-mid-late S by EdU incorporation after the release

Unfortunately, as mentioned above the lack of synchrony, from a release from a contact inhibition arrest, masks synchronous progression through S phase of a RPE1 cell population. We have therefore established EdU incorporation in asynchronous cells, Fig 2a.

13. Figure 3a. The authors should confirm their findings by investigating the level of PCNA by immunoblotting in the same experimental settings.

We have added this analysis in Supplementary Figs 2d (asynchronous cells) and 3a (synchronised cells).

14. Figure 3b. Interestingly, the authors found that the fork length was similar in siControl cells in early and mid/late S phase, (the average length in these cells was ~16 um). In the Figure 2c, for siControl the average length in non-synchronized cells was 17.8. Is this difference due to technical problems of the DNA fibres or due to synchronization? Fibre results from equivalent experiments should be presented and the number of scored fibres for each experiment should

be increased.

We report the additional two biological repeats in the supplementary Fig 3c, left and central panel. Notwithstanding experimental variation, the extent of the increase in DNA fibre length upon E2F6 depletion at 24h is very reproducible.

15. Figure 3c. Is the fork length significantly different between early vs mid/late S, 17.2 vs 18.6? Any explanation why are these numbers different from the ones presented in the Figure 3b for siControl untreated cells?

As discussed above, there is inherently variability in DNA fibre analysis data. Here, we report differences between control and treated samples within the same experiment, which is very reproducible.

16. Figure 4. gH2AX and RPA intensities can be correlated with DNA content. The cell cycle profiles from generations 0, 1, 2 should be shown. I speculate that the cells start to arrest in G1 due to DNA damage.

A quantification of G1 arrest has been added to Fig 4, and the text:

” Single cell time-lapse analysis, using quantification of PCNA pattern and intensity as shown in Figure 1d, **shows a significant increase, in the proportion of arrested cells and in G1 length** in cells treated with siE2F6 (Fig. 4b,c, Supplementary Fig. 4c,d).”

Minor comments:

- It probably is just semantic but the conclusion from results presented in the Figure 1 are overstated (Page 4, the end of paragraph 1). A more precise conclusion should be – kd-E2F6 cells spend less time in S phase.

Whilst we agree that the conclusion drawn by the reviewer (kd-E2F6 cells spend less time in S phase) is of course also correct, as mentioned before, the aim of the study was to establish the effect of increased E2F-dependent transcription in S phase. The title reflects the generality of the effect on E2F-dependent transcript levels in S phase rather than the specific approach we've taken.

- In line 106, incomplete sentence: “This allowed us to monitor.”

We have amended this mistake.

Reviewer #2 (Remarks to the Author):

The study by Pennycook et al. analyzes the role of E2F in controlling S phase length by determining what the authors term “replication capacity”. Replication capacity is a novel and useful parameter of a cell's DNA synthesis kinetics. The

authors essentially demonstrate that E2F regulates the expression of S phase factors and actively adjusts those expression levels to ensure completion of genome duplication with a given number of active replication origins. Importantly, the study modulates E2F6, which represses E2F-driven transcription of S phase proteins in late S phase. The authors show convincingly that E2F6 knockdown accelerates S phase progression in 2 different (a transformed and nontransformed) cell lines by increasing fork speed in late S phase. The data is well-controlled, statistics are solid and the conclusions are well justified. Previous studies have shown that an increase in fork speed is associated with DNA damage. The authors show here that this is not necessarily a direct effect. This is an important observation. The finding that low level damage can be observed in the second cell-cycle after E2F6 knockdown is theoretically a consequence of diminished origin licensing, which would explain why p21 expression is increased and G1 is extended in the KD cells. This has not been explored rigorously but could be easily done by analyzing Mcm2 and Mcm7 chromatin association at the end of G1 of the second cycle. Examining this point would make this particular section of the paper clearer.

Overall, this is a rigorous and important study.

We thank the reviewer for their positive comments on our study. We agree that it would be informative, regarding the increase in p21 levels in G1 in the second cycle, to be able to assess origin licensing in this phase. However, RPE1 cells are hard to arrest and even harder to synchronously release, with many cells releasing from the arrest at much later times. It is therefore hard, to impossible, to assess Mcm chromatin association in G1 specifically in the second cycle. However, we believe that our data already suggests that a diminished origin licensing in the second cycle is unlikely. Since E2F6 is an E2F target that only accumulates during S phase E2F6 knock-down will have no effect in the first G1 of a cell cycle. After 24h of E2F6 depletion (Figure 2b) a significant proportion of cells in G1 are in the second cycle. Therefore, if there is a diminished origin licensing in the second cycle we should expect a decrease in Mcm binding, which we don't observe. Whilst we acknowledge that this is not the ideal experimental setup we hope the reviewer agrees that it does suggest that it is unlikely that diminished origin licensing in the second cycle is at the basis of the increased p21 levels.

Reviewer #3 (Remarks to the Author):

In this manuscript Pennycook et al. explore the impact of E2F transcriptional activity de-repression in S-phase dynamics of human cells. Authors observe that increasing E2F transcription by interference of the repressor E2F6 results in a faster S-phase completion. A slight increase in EdU incorporation is observed in E2F6 interfered cells, which correlates with longer replication tracks in DNA fibers, suggesting that enhanced E2F transcription increases fork speed. Conversely, overexpression of E2F6 in S-phase reduces replicated fiber length

and EdU incorporation. Lastly, a slight increase is observed in gH2AX phosphorylation and RPA2 intensity, proxies of DNA damage signaling and replication stress, respectively, in E2F6 interfered cells, which correlate with an increase in p21 foci and slight lengthening in G1 duration. Based on these results authors propose that E2F controls the replication capacity of the cell (defined by fork speed and number of active forks), as part of a mechanism that would respond to fluctuations in replication initiation events to maintain absolute synthesis rates and S-phase duration.

The data presented are in general of good quality and the conclusions are original and of potential interest for researchers in the replication and cell cycle fields. However, there are some issues about experimental design and data interpretation raising concerns that should be addressed to support the main conclusions and for the model to be convincing.

Major points:

- Authors define the concept of a replication capacity, which should be kept somewhat constant to promote optimal S-phase length and replication speed, and propose that E2F transcription works within a mechanism controlling replication capacity. Replication rates and fork speed seem increased upon E2F6 interference. However, to conclude that replication capacity is enhanced by E2F transcription in this context authors should prove that the frequency of initiation events is unchanged. This should be straight forward, as origin firing frequencies could be measured in the double label fiber experiments shown. MCM7 data on figure 2b seem to be an estimate of nuclear MCM levels rather than a measure of origin licensing/firing.

We thank the reviewer for this suggestion and have now added measurements of origin firing frequencies from the DNA fibre experiments (Supplementary Fig 2e asynchronous cells, 3d synchronised cells). This analysis shows there is no difference between samples from control and siE2F6 treated cells, suggesting that the frequency of initiation events is unchanged. The data has been added (Supplementary Fig 2e and 3d) and the following sentences added to the text:

“In agreement with this finding, the amount of origin firing observed in the DNA fiber spreads did not change upon E2F6 depletion (Supplementary Fig 2e).”

“This is not coupled with a change in origin firing (Supplementary Fig. 3d).”

- Authors should provide evidence that E2F transcription responds to changes in replication initiation event numbers to support the notion that it acts within a mechanism regulating replication capacity.

Our data shows that changes in the levels of E2F-dependent transcription during

S phase affects the DNA synthesis rate and thereby S phase length indicating E2F-dependent transcription sets the replication capacity of the cell. By setting the replication capacity, E2F-dependent transcription provides a mechanism to ensure timely completion of DNA replication which is independent of potential fluctuations of replication initiation events. We therefore believe that E2F-dependent transcription does not respond to any changes in replication initiation events, but controls the global amount of replication factors, limiting maximum DNA synthesis level, to determine the replication capacity of the cell.

- A key point is which E2F targets promote an enhancement of replication fork speed. A straightforward mechanism would be upregulating dNTP pools via induction of ribonucleotide reductase expression. Authors should measure dNTP levels in control and E2F6 interfered cells to rule out this possibility.

This is a very good point, which we have considered, but don't believe that merely increasing dNTPs pools could enhance replication fork speed as observed in siE2F6 treated cells. It has been shown that the addition of dNTPs can rescue DNA replication fork speed in cells experiencing oncogene-induced replication stress. However, this is in conditions where replication fork speed has been slowed down or stalled. In contrast, several studies report that increasing dNTP availability on its own, in cells not experiencing replication stress, does not increase replication fork speed in human cells (Bester et al. 2011, Saldivar 2012, Técher et al. 2016). We have confirmed this in our system where nucleoside addition is unable to increase fibre length.

Based on this we believe that upregulation of dNTP pools alone is unlikely to be the cause of the fork speed increase in our system, but that this requires upregulation of many E2F targets. In line with this, we observed a clear increase in many replication factors in the E2F6-depleted cells compared to control, including Pol α , Cdc45, Cyclin E and A, Cdc7 MCM2 and PCNA.

Minor point:

- DNA damage signaling and replication stress are addressed 21 hours after cell cycle block release when a small proportion of cells (5-7%) is in S-phase. These experiments would be more informative if performed at later time points when more cells have ongoing replication.

The dot plot (Figure 4a) shows that the % of cells with ongoing replication at 21 hours is much higher than the FACS data suggests (Supplementary figure 3). It shows that a large proportion of cells at the 21-hour time point already have higher RPA signal when pre-extracted (represented as orange dots) indicating that about half the population have ongoing replication at this point. The underestimation of S phase cells at 21 hours, by FACS, is likely due to early S phase cells showing a close to 2N DNA content.

Reviewers' comments:

Reviewer #1 (Remarks to the Author):

In the revised manuscript, the authors have made a good effort to answer my previous comments. On the other hand, I still believe that the differences in the length of S phase (around 10% faster in siE2F6) are too small and such modest differences can be the result of experimental variation. Also, the difference of fork speed after siE2F6, around 10%, is too small to make any solid conclusions along the lines the authors put forward.

Below, I specifically comment on the issues that still remain to be convincingly addressed, in the interest of authors' own reputation and providing solid and well supported evidence. Hence, before suggesting the acceptance of their manuscript, the authors should show convincingly that E2F6-depletion indeed does accelerate fork speed during mid/late S phase. This is an important part of the mechanism to explain how low-level E2F6 reduces the length of S phase without altering origin activity. At this stage, I am not satisfied with their answers to the following related questions raised in the original submission.

Specifically,

11. (original concern): Why do siE2F6a-treated cells show no faster S phase after release from contact inhibition (Suppl. Fig 3), but faster replication forks? This contradicts the main message of the paper. It rather seems that contact inhibition and serum starvation have different impact on S phase length after E2F6-depletion despite increased fork speed, which is not addressed by the authors (Fig. 1 and Suppl Fig. 1 in contrast to Suppl. Fig. 3).

This study is indicative:

Yang HW, Chung M, Kudo T, Meyer T. Competing memories of mitogen and p53 signalling control cell-cycle entry. *Nature*. 2017 Sep 21;549(7672):404-408. doi: 10.1038/nature23880. Epub 2017 Sep 6. PubMed PMID: 28869970.

Authors response: We agree with the reviewer that the effect on S phase length, seen in the single cell time lapse analysis, is not obvious in a synchronized RPE1 cell population. We believe that this is due to the lack of a robust synchronous release with RPE1 cells, which probably masks the decrease in S phase length in this setup. RPE1 cells are hard to arrest and synchronously release. Whilst the enrichment in the S phase population does allow us to observe an increase in DNA fibre length in middle/late S phase the extent of the synchrony is insufficient to evaluate S phase length. It is therefore, unfortunately, not possible to draw any conclusions about S phase length from this data.

My new comment on the rebuttal in this point:

Therefore it is not possible to draw any conclusion about different stages of S phase and correlate those with increased fork speed in mid/late S phase due to the lack of robust synchrony that the authors are experiencing with this model.

Authors could evaluate fork speed using synchronised T98G cells +/-siE2F6 to test fork speed at mid/late S phase.

Alternatively, following the logic of the results presented in Fig 3, authors could test the length of S phase in RPE1 cells that have been manipulated to overexpress E2F6.

12. Original concern: Also, the cell cycle profiles in the Figure S3b at 24 h look similar between siControl and siE2F6. The authors should quantify the amount of cells in early-mid-late S by EdU incorporation after the release

Authors' response: Unfortunately, as mentioned above the lack of synchrony, from a release form

a contact inhibition arrest, masks synchronous progression through S phase of a RPE1 cell population. We have therefore established EdU incorporation in asynchronous cells, Fig 2a.

My new comment on the rebuttal in this point:

This just further supports my previous serious concern and the fact that the data is still inconclusive.

13. Original concern: Figure 3a. The authors should confirm their findings by investigating the level of PCNA by immunoblotting in the same experimental settings.

Authors' response: We have added this analysis in Supplementary Figs 2d (asynchronous cells) and 3a (synchronised cells).

My new related comment on the rebuttal in this point:

Authors could analyse and plot the PCNA mean intensity vs DNA content using high-content image analysis. This will show directly that cells in mid/late S phase contain higher level of PCNA. In Cimprich et al, (Science 2018) there are examples of this type of analysis using EdU vs DNA content. In Pennycook's manuscript, authors should pre-extract cells to quantify PCNA associated to chromatin and correlate the level with the DNA content +/-siE2F6.

Other comments were addressed adequately.

Reviewer #2 (Remarks to the Author):

I am utterly confused by the response to my request to analyze the status of Mcm2-7 binding in the second G1 phase (Fig. 4). The authors basically argue that the cells are no longer synchronized. However, they don't say this in the results section of their manuscript, and claim that the second G1 phase is longer (Fig. 4c). They can't have it both ways. I don't understand why they didn't attempt to assess Mcm2-7 loading dynamics in the second G1. If cells are labeled in parallel by Edu and DAPI, then the portion of cells in G1 can be clearly quantified and Mcm2-7 loading can be determined. The authors claim in their response that "Therefore, if there is a diminished origin

licensing in the second cycle we should expect a decrease in Mcm binding, which we don't observe." Where is the data that shows that Mcm2-7 binding is not decreased? That's the very piece of data I was asking for. The fact that p21 expression is increased and G1 is longer in the granddaughters clearly points to a potential licensing defect in the second G1 phase. As I said in my original review, this part of the study needs to be further corroborated. Understanding why DNA damage occurs in the 2nd cell cycle is critical to understanding the consequences of changes in S phase dynamics.

Reviewer #3 (Remarks to the Author):

While authors show that replication dynamics change upon alteration of E2F transcription, in order to claim that E2F determines "replication capacity" to regulate S-phase length authors should demonstrate that E2F transcription actually responds to changes in replication dynamics (e.g. origin firing frequency). In the reviewed manuscript authors have not addressed this major concern, nor they have investigated whether regulation of dNTP levels is a key driver for the observed changes of replication dynamics upon manipulation of E2F transcription (as opposed to a general increase in replication factors postulated to be limiting for replication speed). Hence, the

data presented fail to provide strong support to the manuscript's central conclusion. In order for the study to be suitable for publication in Nature Communications authors should provide strong evidence that E2F transcription actually dynamically responds to replication changes (origin firing or fork speed) to maintain "replication capacity" constant or strictly limit their conclusions to the impact of E2F transcription on replication fork speed throughout the manuscript.

Thank you for considering our manuscript: (MS# NCOMMS-19-06099B-Z) - "**E2F-dependent transcription determines replication capacity and S phase length**". We appreciate the positive response to our initial submission from the reviewers, and we thank them for taking the time to consider our revised version carefully. We have taken their remaining and new suggestions, to improve the paper very seriously, and have performed a number of additional experiments that we feel adequately address their concerns. We have made adjustments to the figures and text, and provide a point-by-point response below. In addition we have revised 'E2F-activity' to 'E2F-dependent transcription' in the title to more accurately reflect our findings and have added a new co-author who was involved in generating the new data.

Reviewers' comments:

Reviewer #1 (Remarks to the Author):

In the revised manuscript, the authors have made a good effort to answer my previous comments

On the other hand, I still believe that the differences in the length of S phase (around 10% faster in siE2F6) are too small and such modest differences can be the result of experimental variation. Also, the difference of fork speed after siE2F6, around 10%, is too small to make any solid conclusions along the lines the authors put forward.

We would like to thank the reviewer for their positive remarks regarding our effort to answer their previous questions. With regards to the new issue raised by the reviewer, the 10% difference in both S phase and fiber length is consistently observed in different experimental setups and in multiple repeats. This strongly suggests that experimental variation is unlikely at the basis of the observed difference.

Below, I specifically comment on the issues that still remain to be convincingly addressed, in the interest of authors' own reputation and providing solid and well supported evidence. Hence, before suggesting the acceptance of their manuscript, the authors should show convincingly that E2F6-depletion indeed does accelerate fork speed during mid/late S phase. This is an important part of the mechanism to explain how low-level E2F6 reduces the length of S phase without altering origin activity. At this stage, I am not satisfied with their answers to the following related questions raised in the original submission.

We agree with the reviewer that this is an important point and we have carried out the suggested experiments. We have now included new data, which is included in Supplementary Fig. 3d, that shows accelerated fork speed during mid/late S phase in synchronized T98G cells. We hope that this additional data adds solid evidence to further support our conclusions.

Specifically,

11. (original concern): Why do siE2F6a-treated cells show no faster S phase after release from contact inhibition (Suppl. Fig 3), but faster replication forks? This contradicts the main message of the paper. It rather seems that contact inhibition and serum starvation have different impact on S phase length after E2F6-depletion despite increased fork speed, which is not addressed by the authors (Fig. 1 and Suppl Fig. 1 in contrast to Suppl. Fig. 3).

This study is indicative:

Yang HW, Chung M, Kudo T, Meyer T. Competing memories of mitogen and p53 signalling control cell-cycle entry. *Nature*. 2017 Sep 21;549(7672):404-408. doi: 10.1038/nature23880. Epub 2017 Sep 6. PubMed PMID: 28869970.

Authors response: We agree with the reviewer that the effect on S phase length, seen in the single cell time lapse analysis, is not obvious in a synchronized RPE1 cell population. We believe that this is due to the lack of a robust synchronous release with RPE1 cells, which probably masks the decrease in S phase length in this setup. RPE1 cells are hard to arrest and synchronously release. Whilst the enrichment in the S phase population does allow us to observe an increase in DNA fibre length in middle/late S phase the extent of the synchrony is insufficient to evaluate S phase length. It is therefore, unfortunately, not possible to draw any conclusions about S phase length from this data.

My new comment on the rebuttal in this point:

Therefore it is not possible to draw any conclusion about different stages of S phase and correlate those with increased fork speed in mid/late S phase due to the lack of robust synchrony that the authors are experiencing with this model.

Authors could evaluate fork speed using synchronised T98G cells +/-siE2F6 to test fork speed at mid/late S phase.

Alternatively, following the logic of the results presented in Fig 3, authors could test the length of S phase in RPE1 cells that have been manipulated to overexpress E2F6.

As suggested by the reviewer, and also addressed above, we have now evaluated fork speed using synchronised T98G cells +/-siE2F6 to test fork speed at mid/late S phase. This new data, which is included in Supplementary Fig. 3d, convincingly shows that also for these cells E2F6-depletion accelerates fork speed during mid/late S phase. This confirms our data for RPE1 cells and we agree that this was important data that was missing in our earlier version.

12. Original concern: Also, the cell cycle profiles in the Figure S3b at 24 h look similar between siControl and siE2F6. The authors should quantify the amount of cells in early-mid-late S by EdU incorporation after the release

Authors' response: Unfortunately, as mentioned above the lack of synchrony, from a

release from a contact inhibition arrest, masks synchronous progression through S phase of a RPE1 cell population. We have therefore established EdU incorporation in asynchronous cells, Fig 2a.

My new comment on the rebuttal in this point:

This just further supports my previous serious concern and the fact that the data is still inconclusive.

13. Original concern: Figure 3a. The authors should confirm their findings by investigating the level of PCNA by immunoblotting in the same experimental settings.

Authors' response: We have added this analysis in Supplementary Figs 2d (asynchronous cells) and 3a (synchronised cells).

My new related comment on the rebuttal in this point:

Authors could analyse and plot the PCNA mean intensity vs DNA content using high-content image analysis. This will show directly that cells in mid/late S phase contain higher level of PCNA. In Cimprich et al, (Science 2018) there are examples of this type of analysis using EdU vs DNA content. In Pennycook's manuscript, authors should pre-extract cells to quantify PCNA associated to chromatin and correlate the level with the DNA content +/-siE2F6.

As pointed out by the reviewer we addressed their concern, by establishing the levels of PCNA by immunoblotting PCNA in the same experimental settings, in the previous revised version, but now also confirm these results, using immunofluorescence shown in Supplementary Fig. 4, to confirm our previous findings.

Other comments were addressed adequately.

Reviewer #2 (Remarks to the Author):

I am utterly confused by the response to my request to analyze the status of Mcm2-7 binding in the second G1 phase (Fig. 4). The authors basically argue that the cells are no longer synchronized. However, they don't say this in the results section of their manuscript, and claim that the second G1 phase is longer (Fig. 4c). They can't have it both ways. I don't understand why they didn't attempt to assess Mcm2-7 loading dynamics in the second G1. If cells are labeled in parallel by Edu and DAPI, then the portion of cells in G1 can be clearly quantified and Mcm2-7 loading can be determined.

The authors claim in their response that "Therefore, if there is a diminished origin licensing in the second cycle we should expect a decrease in Mcm binding, which we don't observe." Where is the data that shows that Mcm2-7 binding is not decreased? That's the very piece of data I was asking for. The fact that p21 expression is increased and G1 is longer in the granddaughters clearly points to a potential licensing defect in the second G1 phase. As I said in my original review, this part of the study needs to be further corroborated. Understanding why DNA damage occurs in the 2nd cell cycle is critical to understanding the consequences of changes in S phase dynamics.

We apologize to the reviewer for focusing too much on losing some synchrony in the second cycle and not appreciating how we could still use the experimental setup to analyze the status of Mcm2-7 binding in the second G1 phase. We have now established Mcm7 binding by immunofluorescence at 41h after release from confluency (second cycle) for the G1, S and G2/M populations. This data, shown in in Supplementary Figure 4f, indicates there is no reduction, if anything there is an increase, in Mcm7 signal on chromatin, suggesting there is no reduction in licensing in the second cycle.

Reviewer #3 (Remarks to the Author):

While authors show that replication dynamics change upon alteration of E2F transcription, in order to claim that E2F determines "replication capacity" to regulate S-phase length authors should demonstrate that E2F transcription actually responds to changes in replication dynamics (e.g. origin firing frequency). In the reviewed manuscript authors have not addressed this major concern, nor they have investigated whether regulation of dNTP levels is a key driver for the observed changes of replication dynamics upon manipulation of E2F transcription (as opposed to a general increase in replication factors postulated to be limiting for replication speed). Hence, the data presented fail to provide strong support to the manuscript's central conclusion. In order for the study to be suitable for publication in Nature Communications authors should provide strong evidence that E2F transcription actually dynamically responds to replication changes (origin firing or fork speed) to maintain "replication capacity" constant or strictly limit their conclusions to the impact of E2F transcription on replication fork speed throughout the manuscript.

We apologize to the reviewer for not clearly explaining our concept. We do not claim that E2F transcription responds to replication dynamics. Our data suggests that E2F activity determines the replication capacity during S phase, independent of replication dynamics (origin firing or fork speed). Our data suggests that E2F transcription limits how much DNA can be synthesized at any given time during S phase, determining the replication capacity. The replication capacity affects the speed replication forks can travel given a set number of active forks. Because the replication capacity, set by E2F transcription, is limiting a lower number of active forks can travel faster, whilst more active forks result in slower fork speed. We have now included text that clearly explains

that our model does not suggest that changing replication dynamic affects E2F transcription, but that changing E2F transcription does change replication rate as a whole.

To address the reviewers' other concern, related to whether regulation of dNTP levels is a key driver for the observed changes of replication dynamics, we include data below that shows that adding nucleosides to RPE1 cells does not increase DNA fiber length.

RPE1 cells were treated with exogenous nucleosides at the indicated concentrations 48 h prior to DNA fibre analysis. Significance was determined by the Kruskal-Wallis test. Number of fibres counted in each sample shown.

These data, which has also been reported by others whose results we mention in the discussion, suggest that an increase of dNTP levels alone is unlikely the key driver of the observed changes of replication dynamics.

We hope our explanation and additional data addresses the concerns raised by the reviewers.

REVIEWERS' COMMENTS:

Reviewer #1 (Remarks to the Author):

I am now satisfied by the newly added data and authors' responses, and recommend this manuscript for acceptance

Reviewer #2 (Remarks to the Author):

I am now fully satisfied with the revisions. The authors have answered all of my questions.

Anja Katrin Bielinsky

Reviewer #3 (Remarks to the Author):

Throughout the text authors convey the idea that E2F-dependent transcription controls the replication capacity of cells as part of a regulatory mechanism to ensure timely replication completion and to optimise replication fork speed:

P1. "Thus, at its most fundamental level, E2F-dependent transcription determines the replication capacity of a cell during S phase, controlling the time it takes to duplicate the genome. More generally, our work shows that controlling replication capacity during S phase is a robust mechanism to ensure timely completion of DNA replication, which could be at the basis of keeping replication fork speed within an optimal range independently of fluctuations in replication initiation events.

P2. "Limiting the replication capacity of a cell would provide an elegant mechanism to regulate the global rate of replication during S phase, largely independent of the number of active replication forks."

P6. "This suggests that E2F-dependent transcription controls the amount of DNA a cell can synthesis per unit time in S phase, which we define as replication capacity, likely through regulating the expression of limiting factors for DNA replication."

P9. "Overall our work suggests that the tight regulation of E2F transcription in S phase is an important determinant in DNA replication control and genomic stability, which has important implications for basic biology and the understanding of cancer."

However, evidence is not provided on what cues would this mechanism respond to in order to be effective, how E2F-dependent transcription would be modulated to achieve this regulation, or through what factors would replication speed be controlled. Since the expression of many factors important for replication depends on E2F-dependent transcription, it is a logical expectation that increasing E2F-dependent transcription might augment replication rates and the overall replication capacity. The authors nicely show this in this manuscript and demonstrate this preferentially occurs through fork speed and not through increased origin firing, but the notion that E2F-transcription is part of a mechanism controlling replication rates to preserve genome stability remains premature at this point. Regardless the significant amount of work reported in the manuscript, in my opinion evidence supporting these claims is required to firmly support its publication in Nature Communications.

Rebuttal to reviewer #3 comments.

Throughout the text authors convey the idea that E2F-dependent transcription controls the replication capacity of cells as part of a regulatory mechanism to ensure timely replication completion and to optimise replication fork speed:

However, evidence is not provided on what cues would this mechanism respond to in order to be effective, how E2F-dependent transcription would be modulated to achieve this regulation, or through what factors would replication speed be controlled. Since the expression of many factors important for replication depends on E2F-dependent transcription, it is a logical expectation that increasing E2F-dependent transcription might augment replication rates and the overall replication capacity. The authors nicely show this in this manuscript and demonstrate this preferentially occurs through fork speed and not through increased origin firing, but the notion that E2F-transcription is part of a mechanism controlling replication rates to preserve genome stability remains premature at this point. Regardless the significant amount of work reported in the manuscript, in my opinion evidence supporting these claims is required to firmly support its publication in Nature Communications.

We would like to thank the reviewer for their helpful comments and positive assessment of our work. Please find below our response in bold, and the changes we have made to the text in bold underlined, to ensure our conclusions better reflect the data presented.

P1. “Thus, at its most fundamental level, E2F-dependent transcription determines the replication capacity of a cell during S phase, controlling the time it takes to duplicate the genome. More generally, our work shows that controlling replication capacity during S phase is a robust mechanism to ensure timely completion of DNA replication, which could be at the basis of keeping replication fork speed within an optimal range independently of fluctuations in replication initiation events.

We have replaced this text with ***“Thus, E2F-dependent transcription determines the DNA replication capacity of a cell, which affects the replication rate, controlling the time it takes to duplicate the genome and complete S-phase.”***, which removes the text that suggests that *‘E2F-transcription is part of a mechanism controlling replication rates to preserve genome stability’* which reviewer 3 rightly points out *‘remains premature at this point’*.

P2. “Limiting the replication capacity of a cell would provide an elegant mechanism to regulate the global rate of replication during S phase, largely independent of the number of active replication forks.”

We agree with the reviewer that *‘through what factors would replication speed be controlled’* remains unknown and, at this point, this is speculative. To highlight this we have added ***‘We speculate that limiting the replication capacity of a cell would provide an elegant mechanism to regulate the global rate of replication during S phase, largely independent of the number of active replication forks.’***

P6. “This suggests that E2F-dependent transcription controls the amount of DNA a cell can synthesis per unit time in S phase, which we define as replication capacity, likely through regulating the expression of limiting factors for DNA replication.”

We agree with the reviewer that our data does not provide evidence *“on how E2F-dependent transcription would be modulated to achieve this regulation, or through what factors would replication speed be controlled.”*. To reflect this we have changed the sentence to ***“This suggests that E2F-dependent transcription controls the amount of DNA a cell can synthesis per unit time in S phase, defined as replication capacity, which we speculate is likely through regulating the expression of limiting factors for DNA replication”***.

P9. "Overall our work suggests that the tight regulation of E2F transcription in S phase is an important determinant in DNA replication control and genomic stability, which has important implications for basic biology and the understanding of cancer."

We agree with the reviewer that *"the notion that E2F-transcription is part of a mechanism controlling replication rates to preserve genome stability remains premature at this point"*. **To reflect this better we have changed the sentence as follows - Overall our work suggests that the tight regulation of E2F transcription in S phase is an important determinant in DNA replication control. Further research is required to establish a potential role for this regulation in maintaining genomic stability, which would have important implications for basic biology and the understanding of cancer.**